# Is a higher altitude associated with shorter survival among at-risk neonates?

Iván Dueñas-Espín[1]*, Luciana Armijos-Acurio[1], Estefanía Espín[2], Fernando Espinosa-Herrera[2,3], Ruth Jimbo[1], Ángela León-Cáceres[1,4,5], Raif Nasre-Nasser[6], María F. Rivadeneira[1], David Rojas-Rueda[7], Laura Ruiz-Cedeño[8], Betzabé Tello[1], Daniela Vásconez-Romero[8]

1 Instituto de Salud Pública, Facultad de Medicina, Pontificia Universidad Católica del Ecuador (PUCE), Quito, Ecuador, 2 Escuela de Medicina, Facultad de Ciencias de la Salud, Universidad de las Américas (UDLA) and Facultad de Medicina, Universidad Central del Ecuador, Quito, Ecuador, 3 Sociedad Ecuatoriana de Medicina Familiar (SEMF), Quito, Ecuador, 4 Heidelberg Institute of Global Health, Faculty of Medicine, University of Heidelberg, Heidelberg, Germany, 5 Tropical Herping, Quito, Ecuador, 6 Programa de Pós-graduação em Ciências Fisiológicas, Instituto de Ciências Biológicas, Universidade Federal do Rio Grande (FURG), Rio Grande, Brasil, 7 Department of Environmental and Radiological Health Sciences, Colorado State University, Fort Collins, Colorado, United States of America, 8 Postgrado de Pediatría, Facultad de Medicina, Pontificia Universidad Católica del Ecuador (PUCE), Quito, Ecuador

* igduenase@puce.edu.ec

**Data Availability Statement:** All relevant data are within the manuscript and its Supporting information files.

## Abstract

### Introduction

We hypothesize that high altitudes could have an adverse effect on neonatal health outcomes, especially among at-risk neonates. The current study aims to assess the association between higher altitudes on survival time among at-risk neonates.

### Methods

Retrospective survival analysis. Setting: Ecuadorian neonates who died at ≤28 days of life. Patients: We analyzed the nationwide dataset of neonatal deaths from the Surveillance System of Neonatal Mortality of the Ministry of Public Health of Ecuador, registered from 126 public and private health care facilities, between January 2014 to September 2017. Main outcome measures: We retrospectively reviewed 3016 patients. We performed a survival analysis by setting the survival time in days as the primary outcome and fixed and mixed-effects Cox proportional hazards models to estimate hazard ratios (HR) for each altitude *stratum* of each one of the health care facilities in which those neonates were attended, adjusting by individual variables (*i.e.*, birth weight, gestational age at birth, Apgar scale at 5 minutes, and comorbidities); and contextual variables (*i.e.*, administrative planning areas, type of health care facility, and level of care).

### Results

Altitudes of health care facilities ranging from 80 to <2500 m, 2500 to <2750m, and ≥2750 m were associated respectively with 20% (95% CI: 1% to 44%), 32% (95% CI:<1% to 79%) and 37% (95% CI: 8% to 75%) increased HR; compared with altitudes at <80 m.

**Funding:** This work was part of the "Score Bebé" project. IDE, LAA, RJ, MFR, BT were supported by Pontificia Universidad Católica del Ecuador (Pontifical Catholic University of Ecuador, https://www.puce.edu.ec/) grant number QINV0025-IINV533010100. The funders had no role in study design, data collection and analysis, decision to publish, or preparation of the manuscript.

**Competing interests:** The authors have declared that no competing interests exist.

## Conclusion

Higher altitudes are independently associated with shorter survival time, as measured by days among at-risk neonates. Altitude should be considered when assessing the risk of having negative health outcomes during neonatal period.

## Introduction

It is estimated that over 140 million people inhabit an altitude of >2500 m worldwide [1]. They are exposed to an environment which is characterized by the lack of oxygen, intense solar radiation, low temperatures, low humidity, and less fertile soils compared with lower altitudes; moreover, it is a geographically dispersed population with high inequities, predisposing them to negative health outcomes, more so in children [2].

Among the global causes of death in children under 5 years of age, neonatal death accounts for 45.1% [3]. This is influenced by several hierarchical factors; distal ones such as social determinants (*e.g.*, Gini coefficient, poverty) [4, 5], to intermediate and proximal factors (*e.g.*, maternal educational level, coverage of maternal health) [6, 7], and to perinatal health services (*e.g.*, antenatal care, skilled birth attendance) [8].

Scientific evidence has consistently shown that high altitude is associated with high neonatal mortality [9]. Specifically, in the prenatal phase, a pregnancy at high altitude has lower availability of iron, vitamins A and D, iodine [1], higher erythropoietic demands, reduced blood oxygen content [9], and a subsequent lower maternal oxygen delivery to the fetoplacental unit. These mechanisms could result in lower fetal growth [9] and in a decrease in placental weight, accompanied by a significant increase in the incidence of syncytial nodes, trophoblastic cells, and fetal capillaries with a large intervillous space and fewer villi. The disadvantages in oxygen diffusion can also contribute to low birth weight in full-term neonates, preeclampsia, and gestational hypertension [10]. Hence, the transition from oxygenation through the placenta to oxygenation through the lungs is poor in high altitudes [9] the transition period for neonates in high altitudes is longer; it means that, even in the absence of pulmonary disorders, neonates may experience arterial oxygen desaturation during the first week of life [9].

Specifically, in the prenatal phase, a pregnancy at high altitude has lower availability of iron, vitamins A and D, iodine [1], higher erythropoietic demands, reduced blood oxygen content [9], and a subsequent lower maternal oxygen delivery to the fetoplacental unit. These mechanisms could result in lower fetal growth [9] and in a decrease in placental weight, accompanied by a significant increase in the incidence of syncytial nodes, trophoblastic cells, and fetal capillaries with a large intervillous space and fewer *villi*. The disadvantages in oxygen diffusion can also contribute to low birth weight in full-term neonates, preeclampsia, and gestational hypertension [10]. Hence, the transition from oxygenation through the placenta to oxygenation through the lungs is poor in high altitudes [9] the transition period for neonates in high altitudes is longer; it means that, even in the absence of pulmonary disorders, neonates may experience arterial oxygen desaturation during the first week of life [9].

However, past studies have an ecological approach [11–14], small sample sizes [15], lack of consideration of individual confounders in the analyses (i.e., birth weight, gestational age, small for gestational age, Apgar scale, type of delivery, and the presence of specific comorbidities) [15], lack of contextual factors (i.e., differences in quality of care, type of health care facility [i.e., public or private], and level of care [16–18], and most of them only compared two or three altitude categories [11–15, 19–21]. Despite the available conceptual model of stillbirth

health determinants [5, 22], this model does not include altitude as a determinant. This background represents an evidence-based conceptual framework that suggests that neonatal survival could be negatively associated with high altitude, especially among at-risk neonates. The current study aims to assess such association by setting survival among at-risk neonates, as measured by days, as the primary outcome, using a nationwide information database.

## Materials and methods

See the complete version in the S1 File.

### Design

We performed a nationwide retrospective analysis of registered deaths in Ecuador of all neonates who died at ≤28 days of life and were registered in 126 public and private health care facilities into the Surveillance System of Neonatal Mortality of the Ministry of Public Health of Ecuador, from January 2014 to September 2017. We excluded from the analyses neonates who died in a prehospital setting due to lack of information about the altitude of their health care attendance S1 Fig.

### Population and database

All registered neonatal deaths were included in this study. The epidemiologist in each health care facility systematically reported the cases to this system; describing neonatal deaths up to 24 hours after they happened. The database includes perinatal information from all registered deceased neonates—at-risk neonates—but it did not include information on maternal education and other contextual variables at the individual level.

### Main outcome

We performed a survival analysis by setting the neonatal survival time, as measured by days, as the primary outcome. Then, we calculated hazard ratios (HR) using Cox's proportional hazards models and we compared the survival time across altitude strata [23]. Despite information about non-deceased neonates being unavailable, the survival analysis gave us the probability of event occurrence (*i.e.*, neonatal death) per each altitude stratum.

### Main explanatory variable

The altitude of the health care center in which neonates were attended was the main explanatory variable. Three independent researchers used information from the GeoSalud 3.6.0 web viewer www.geosalud.com and the computer software www.googlearth.com to verify the altitude of each facility. To categorize the altitude, we built a histogram to evaluate the possibility of dividing altitude categories by centiles; but, due to the irregular Ecuadorian geography, the particular distribution of neonatal deaths in each altitude interval made the categorization in tertiles or quartiles very difficult (S2 Fig). However, a continuous analysis of the altitude data could reduce unintended biases and be more useful for comparison of estimates with other studies. Thus, we decided to only keep a categorical analysis due to the important overdispersion of the data (S3 Fig). In order to perform a proper categorization of the altitude we considered that Ecuador has four regions, each with different altitude and a specific distribution of the population [24]. Nevertheless, regarding the health outcome evaluated, distinct studies established different altitude clusters. Thus, the patterns of altitude categorization we used, are according to our research objective (see details in the S1 File).

## Other covariates

Given that there are several potential confounders in the association between altitude and neonatal survival time, variables were divided into two types: *(i)* individual covariates: birth weight, gestational age at birth, small for gestational age—neonates whose birth weight was less than the 10th percentile for gestational age–[25], using the Intergrowth equations [26], Apgar scale at 5 minutes, type of delivery, and comorbidities; and, *(ii)* contextual covariates: administrative planning areas, type of health care facility, and level of care, 2014 to 2016 Gini coefficient at the province level, province of the habitual residence of the mother, as well as location of the health care centres (province), to identify neonates who died outside their mother's usual residence, and rural-urban health care facilities.

## Statistical analyses

Despite the analyses being performed using the whole database of Ecuadorian neonatal mortality from 2014 to 2017, we corroborated that a sample size of no less than 100 observations per altitude stratum was enough to allow meaningful comparative survival analyses of the chosen categories [27] (S1 File). Finally, we analyzed 3016 deceased neonates with 156 patients belonging to the lowest altitude *stratum*.

Descriptive statistics were performed using percentages for categorical variables and median and P25 to P75 for discrete variables. To assess the differences of each individual and contextual variable across altitude categories, we performed: *(i)* Kruskal Wallis tests for assessing differences of Apgar score at 5 minutes and gestational age; *(ii)* for assessing differences in birth weight we used one-way ANOVA and Tukey post hoc test, *(iii)* Chi$^2$ tests for assessing differences of small for gestational age, type of delivery, comorbidities, tertiles of Gini at the provincial level, type of health care center, level of care, neonatal deaths outside their mother's usual residence province, and rural-urban health care facility; and, *(iv)* log-rank test for equality of survivor functions for assessing differences of neonatal survival time, as measured by days,—the main outcome–.

Then, we estimated crude and adjusted HR per each stratum of altitude [23]. In that sense, we built multivariate Cox proportional hazards models to evaluate the independent association between altitude and neonatal survival time among at-risk neonates.

We built a saturated model including the covariates that were selected according to the researcher's expertise. We selected the covariates that were retained in the parsimonious model (S1 Table) which shows the excluded variables from the saturated model, as well as the estimates resulting from the parsimonious model. Hazard ratios were estimated. The 95% confidence intervals (95% CI) of the HR and their corresponding *p-values* were obtained. Once the parsimonious model was obtained, we compared both models and chose the "final" model, according to its level of significance from the likelihood ratio test.

To assess the effects from contextual variables on mortality we estimated HR by mixed-effects Cox proportional hazards models, considering contextual variables in model 1: administrative planning areas, type of health care facility, and level of care, in model 2: administrative planning areas, and level of care; and in model 3: level of care, only.

We performed several secondary analyses to assess the sensitivity of our estimates to our assumptions regarding biases, as well as to test for model misspecification. First, considering that differential treatment for individual causes of death (for example, asphyxia, infectious diseases, etc.) could affect the estimates, we ran a final model excluding neonates who died from *(i)* asphyxia related disorders, *(ii)* congenital malformations, *(iii)* prematurity related disorders, and *(iv)* infectious disorders. Second, to verify that estimations did not change significantly between the highest and lowest risk of adverse outcomes we excluded those neonates with <5

points and those with ≥7 Apgar score at 5 minutes. Third, knowing that migration from the residence where the pregnancy occurred could modify the estimates, we excluded neonates who died in a different province than their mother's habitual residence. Fourth, after categorizing survival in four different binary outcomes: death before 24, 48, 72 hours, and before 7 days of life, we ran four mixed-effects multivariate logistic regression models to estimate odds ratios by adjusting according to model 1. Fifth, regarding the fact that the database is a death registry—and all neonates die–, it represents a right truncated database. In that sense, we performed an Inverse Probability Weighted (IPW) Cox Regression analysis [28, 29] and calculated the estimates of survival in days of life across altitude strata by artificial cut-offs at 15, 17, 20 and 23 days of the retrospective follow-up (S1 File).

Furtherly, to assess the effects from other contextual variables like Gini coefficient at province level and rural-urban health care, we estimated HR from mixed-effects multivariate Cox proportional models in this way: fixed effects for next individual variables: gestational age, birth weight, Apgar scale at 5 minutes, comorbidities, and random effects for contextual variables: *(i)* Gini coefficient at the provincial level, type of health care facility, and level of care in Model 4; *(ii)* Gini coefficient at the provincial level in Model 5; *(iii)* rural-urban health care facility, type of health care facility and level of care in Model 6; and *(ii)* rural-urban health care facility in Model 7.

Considering the small number of missing data (S2 Table) we employed complete case analysis in estimating statistical associations. We considered that there were statistically significant differences across health altitude strata when the *p-value*<0.05; all analyses were performed with Stata 14.2 (Statistical Software Stata: Release 14.2 College Station, TX: StataCorp LP).

## Ethics approval

This study was part of the "Score Bebé" project and was conducted with the authorization of the Research Ethics Committee in Human Beings (CEISH) of the *Pontificia Universidad Católica del Ecuador* approval (code number: 2018-09-EO). The need for consent was waived by the ethics committee given that the study was based on secondary information; therefore, informed consent was not obtained. All data weres fully anonymized before accessed.

## Results

The study identified 3016 neonates who died between January 2014 to September 2017, with a median gestational age (P25 to P75) of 32 (28 to 37) weeks at birth, a median (P25 to P75) birth weight of 1402 (900 to 2400) g, and a median (P25 to P75) Apgar score at 5 minutes of 6 (4 to 8). Except for the percentage of small for gestational age, there were significant differences between the individual and contextual variables across altitude categories (Table 1).

According to the fixed-effects models, and without considering random effects from contextual variables, the highest altitude was significantly associated with shorter neonatal survival time by using two different categorizations: *(i)* considering the reference value of <10 m, being attended between 10 m to <2500 m, and ≥2500 m there was 23% and 26% higher HR, respectively (p for trend <0.01); and, *(ii)* considering being attended at <80 m as the reference, being attended between 80 to <2500 m, from 2500 to <2750 m, and ≥2750 m there was 42%, 57% and 17% higher HR, respectively (*p for trend* = 0.01, Table 2).

After running mixed-effects models which included random effects from contextual variables (1, 2 and 3), estimations corroborated a higher HR as altitude increases. In this way: considering being attended at <80 m as the reference, being attended between 80 to <2500 m, from 2500 to <2750 m, and ≥2750 m there was 20%, 32%, and 37% higher HR in model 1; 23%, 25% and 32% higher HR in model 2; and, 19%, 32%, and 41% higher HR in model 3,

**Table 1. Differences on individual and contextual variables of the study population across categories of altitude of the health facility where neonates were attended.**

| Individual and contextual variables[a] | Altitude of the health facility where neonates were attended. Total n = 3016 | | | | |
|---|---|---|---|---|---|
| | <80 m (n = 1625) | ≥80 to <2500 m (n = 405) | ≥2500 to <2750 m (n = 156) | ≥2750 m (n = 830) | p-value |
| **Individual variables[b]** | | | | | |
| Gestational age in weeks, median (P25 to P75) | 31 (27 to 36) | 33 (28 to 38) | 31 (26 to 35) | 32 (27 to 36) | <0.01 |
| Birth weight in g, median (P25 to P50) | 1633 (915) | 1875 (976) | 1556 (878) | 1634 (908) | <0.01 |
| Small for gestational age, n (%) | 231 (15) | 66 (17) | 28 (19) | 120 (15) | 0.45 |
| Apgar score at five minutes, median (P25 to P75) | 6 (4 to 7) | 7 (5 to 9) | 7 (4 to 9) | 8 (5 to 9) | <0.01 |
| Type of delivery | | | | | |
| *C-section, n (%)* | 1025 (64) | 192 (48) | 52 (34) | 451 (55) | <0.01 |
| *Vaginal delivery, n (%)* | 509 (32) | 187 (47) | 81 (53) | 311 (38) | <0.01 |
| *Dystocic delivery, n (%)* | 66 (4) | 22 (5) | 20 (13) | 62 (8) | <0.01 |
| Comorbidities | | | | | |
| *Asphyxia related disorders, n (%)* | 418 (26) | 123 (30) | 33 (21) | 152 (18) | <0.01 |
| *Malformations, n (%)* | 371 (23) | 92 (23) | 29 (19) | 206 (25) | <0.01 |
| *Prematurity related disorders, n (%)* | 473 (29) | 83 (21) | 60 (38) | 288 (35) | <0.01 |
| *Infectious diseases, n (%)* | 307 (19) | 93 (23) | 29 (19) | 147 (18) | <0.01 |
| *Other non-previously classified, n (%)* | 56 (4) | 14 (4) | 5 (3) | 37 (4) | <0.01 |
| Incidence rates of death per 100 person-days, 95% CI[b] | 16.6 (15.7 to 17.5) | 21.6 (19.6 to 23.8) | 21.5 (18.4 to 25.2) | 17.2 (16.1 to 18.4) | <0.01 |
| **Contextual variables[c]** | | | | | |
| 2014 to 2016 GINI coefficient, mean (SD) | 0.44 (0.03) | 0.49 (0.05) | 0.48 (0.02) | 0.47 (0.02) | <0.01 |
| *First tertile, <0.426, n (%)* | 1087 (68) | 16 (4) | 0 | 1 (<1) | <0.01 |
| *Second tertile, >0.426 to 0.464, n (%)* | 315 (19) | 144 (38) | 84 (54) | 640 (78) | |
| *Third tertile, >0.464, n (%)* | 207 (13) | 225 (58) | 72 (46) | 177 (22) | |
| Type of health care center | | | | | |
| *Private medical care, n (%)* | 721 (44) | 26 (6) | 0 (0) | 62 (7) | <0.01 |
| *Public medical care, n (%)* | 904 (55) | 379 (94) | 156 (100) | 768 (92) | <0.01 |
| Level of care | | | | | |
| *Primary care unit, n (%)* | 8 (<1) | 13 (3) | 5 (3) | 27 (3) | <0.01 |
| *Secondary care unit, n (%)* | 507 (31) | 384 (96) | 150 (97) | 282 (34) | <0.01 |
| *Tertiary care unit, n (%)* | 1107 (68) | 3 (1) | 0 (0) | 517 (63) | <0.01 |
| Neonatal deaths outside their mother's usual residence province | | | | | |
| *Yes, n (%)* | 304 (19) | 56 (14) | 6 (4) | 171 (21) | <0.01 |
| *No, n (%)* | 1321 (82) | 349 (86) | 150 (96) | 659 (79) | |
| Rural-urban health care facility | | | | | |
| *Rural, n (%)* | 511 (59) | 151 (37) | 68 (44) | 135 (16) | <0.01 |
| *Urban, n (%)* | 1114 (52) | 254 (63) | 88 (56) | 695 (84) | |

[a]There were missing data in some variables (S1 Table). Except for the percentage of Small for Gestational Age, there were significant differences of the other variables across altitude categories (p-values <0.01, by Chi2, ANOVA and Tukey post hoc test, Kruskal Wallis; see main text for details).

[b]To assess statistical differences on survival across altitude *strata* we employed the Log-rank test for equality of survivor functions, (see main text for details).

[c]All individual and contextual variables were significantly associated with actual time in days to death among at-risk neonates.

respectively for each stratum (*p for trend* = 0.01) (S1–S3 Tables) which demonstrated a dose-response shape (Fig 1).

Sensitivity analyses performed to assess if the estimates changed with specific exclusions yielded similar results (S4–S10 Tables). Consequently, neither excluding based on individual causes of death nor excluding the lowest and highest risk neonates, nor those who migrated to a different province than their mother's habitual residence, did not change the estimates.

**Table 2.  Crude and adjusted hazard ratios of the incidence rate of death per each altitude stratum, estimated by fixed effects Cox proportional hazards models for two different types of categorization of altitude of the health care facility where neonates were attended.**

| Categories of altitude of the health facility where neonates were attended | n (%) | Crude hazard ratio[a] (95% CI) | p-value | Adjusted hazard ratio[b] (95% CI) | p-value |
|---|---|---|---|---|---|
| **Categorization 1** | | | | | |
| *First tertile, 0 to <11 m (ref.)* | 1024 (34) | 1 | - | 1 | - |
| *Second tertile, ≥11 to <2500 m* | 1006 (33) | 1.11 (1.01 to 1.21) | 0.03 | 1.24 (1.12 to 1.37) | <0.01 |
| *Third tertile, ≥2500 m* | 986 (33) | 1.07 (0.98 to 1.18) | 0.14 | 1.26 (1.14 to 1.40) | <0.01 |
| *p for trend* | - | 1.03 (0.99 to 1.08) | 0.16 | 1.12 (1.06 to 1.18) | <0.01 |
| **Categorization 2** | | | | | |
| *0 to <80 m (ref.)* | 1625 (54) | 1 | - | 1 | - |
| *≥80 to <2500 m* | 405 (13) | 1.22 (1.09 to 1.36) | <0.01 | 1.42 (1.26 to 1.59) | <0.01 |
| *≥2500 to <2750 m* | 156 (5) | 1.26 (1.06 to 1.48) | <0.01 | 1.57 (1.31 to 1.88) | <0.01 |
| *≥2750 m* | 830 (28) | 1.03 (0.94 to 1.19) | 0.56 | 1.17 (1.06 to 1.28) | <0.01 |
| *p for trend* | - | 1.01 (0.98 to 1.04) | 0.39 | 1.05 (1.02 to 1.09) | <0.01 |

[a] Fixed effects univariate Cox proportional hazards model with altitude as the only variable.

[b] Fixed effects multivariate Cox proportional hazard model adjusted by gestational age, birth weight, Apgar scale at five minutes, and comorbidities.

Further, mixed-effects models corroborated that neither Gini coefficient at the province level nor rural-urban health care, affect the estimates (S11 and S12 Tables). When we ran mixed-effects logistic regression models for the four binary outcomes, they yielded similar results, but estimations were non statistically significant in most models, as it was in the Cox proportional

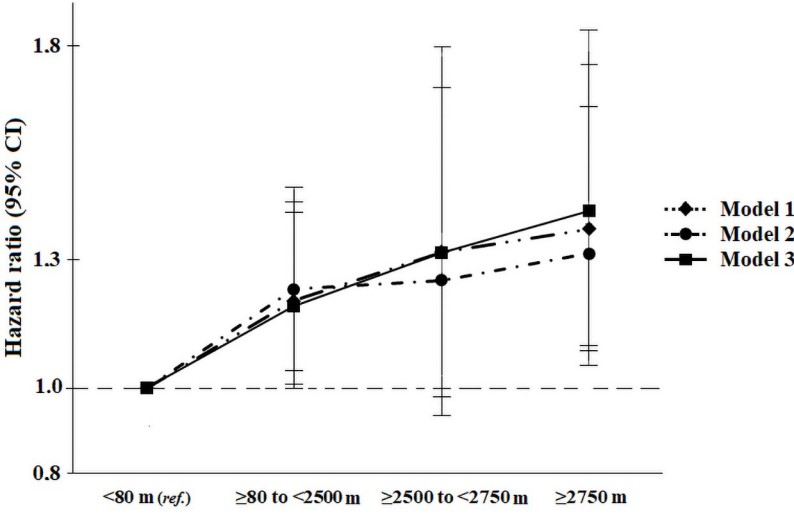

**Fig 1. Adjusted hazard ratios of the incidence rate of death per each altitude stratum according to three mixed-effects multivariate Cox proportional hazards models, considering the reference an altitude of the health care facility where neonates were attended of <80m.** Estimated hazard ratios from mixed-effects multivariate Cox proportional models. All models (1, 2 and 3) estimated fixed effects for next individual variables: gestational age, birth weight, Apgar scale at five minutes, and comorbidities; and random effects for contextual variables in this way: *(i)* administrative planning areas, type of health care facility, and level of care in Model 1, *(ii)* administrative planning areas, and level of care in Model 2; and *(iii)* level of care in Model 3.

hazards modelling (S13 Table). Finally, when performing IPW Cox regression analysis by four different cut-offs of the follow-up, the estimates were in the same direction as the mixed-effects Cox proportional hazards models. Specifically, when the cut-off was at 15 days, estimates resulted in an average (95% CI) survival time of 3.9 (3.7 to 4.1) days of life at <80m, and the difference in survival time was -0.6 (-1.0 to -0.1) days sooner at > = 80 m to <2500 m, -1.2 (-1.9 to -0.5) days sooner at 2500 to <2750; and -0.9 (-1.3 to -0.5) days sooner at ≥2750 m (S14 Table).

## Discussion

We found that the higher the altitude of the health care facility is, where at-risk neonates were attended, the higher their hazard ratio. This association was independent of relevant confounders, even considering contextual variables. Previous studies have suggested an association between altitude and neonatal mortality, but they were based on ecological approaches, small sample sizes, lack of potential confounders, and a comparison of only two altitude categories. This study overcomes such limitations.

Exposure to an increasing altitude induces neonates to develop compensatory mechanisms to improve survival by mechanisms like higher hemoglobin production for increasing oxygen availability in peripheral tissues of the fetus and neonate. Nevertheless, these mechanisms may not be enough to tackle altitude challenges; consequently, we believe that, unfortunately, high altitude overcomes such compensatory mechanisms among at-risk neonates [9]. Despite finding an inverted J shape association between survival and altitude during fixed effects modelling estimations, with an apparent "improvement" in the survival at a certain level (>2750 m) when comparing to lower altitudes (but not to the lowest), several considerations could preclude us from establishing such a conclusion.

First, when accounting for the data clustering (of administrative planning areas, type of health care facility, and level of care), the random-effects modelling provided a useful approach for simultaneously estimating the parameters of the regression model and the variance components that account for the contextual factors. Therefore, it resulted in more realistic estimations.

Second, as previous evidence has suggested, altitude is not only a physical characteristic, but it is composed of complex social, environmental, and healthcare characteristics. For example, Pichincha—the province where the Ecuadorian capital Quito is located (2850 m)–has more health services and public practice physicians per capita than any other province in the country [30]. This could partially explain the reduced HR compared with lesser altitudes (but not with the lowest altitude). Moreover, our results showed that individual and contextual variables were significantly different across altitudes strata. This supports the statement that altitude is not only a challenge in genetic, epigenetic, and physiologic domains, but a complex interaction between the environmental, social, and biological domains.

Therefore, contextual inclusion in the modelling is necessary. Furthermore, we believe that the inverted J shape hypothesis is a relevant issue to explore, but it goes beyond the scope of this study. We believe that the association between altitude and survival is more like a linear shape.

Although the altitude is not a modifiable circumstance for most neonates, it should be considered when assessing the neonatal risk [31]. As it was previously postulated [9], it is required to include effective interventions for further reducing unnecessary child deaths. Neonates born at high altitudes face adverse socio-cultural and physiological challenges compared to those who born at sea levels. We corroborate that altitude is a crucial factor of neonatal health; therefore, it should be considered and registered into a system which monitors pregnant women and neonates, especially at highlands.

Our study overcomes the limitations of previous studies by using a nationwide dataset of at-risk neonates. This was a retrospective survival analysis, using mixed-effects models, providing realistic estimates of the neonatal HR per each altitude stratum. We had enough information for us to adjust according to relevant individual confounders and take into consideration the random effects from contextual variables. Moreover, knowing that altitude at where pregnancy occurred is an important factor that strongly influences the neonatal mortality risk [2], we corroborated that excluding neonates who died outside their mother's usual residence province, did not change the estimates.

Our study had several limitations. First, considering that we did not assess the effects of altitude on low-risk neonates we cannot assure if there is an accentuated effect from altitude on health outcomes in at-risk neonates *vs*. low-risk ones. Nonetheless, a neonatal risk assessment should be complemented by an environmental and social context evaluation for proper stratified clinical management [31].

Second, the present study had a lack of data on individual socioeconomic factors—a strong neonatal health determinant–[3]. However, we believe that the aggregated data used in this analysis could capture such socioeconomic effects, resulting in robust estimations.

Also, when altitude is considered as a potential health determinant in the neonatal period, it is required to consider that the mother-baby binomial could carry genetic traits that are not well suited to their current environment. This might define their degree of adaptation to high altitude. Thus, affect their susceptibility to maladaptation problems that could increase mortality risk [2, 32]. Neonatal mortality at high altitude in South America is reduced by maternal and paternal indigenous Andean ancestry, in a dose-dependent manner [33, 34], mechanisms and genes related to the ancestry protective effect are related to enhanced uteroplacental blood flow [32, 33, 35, 36]. Additionally, despite epigenetic mechanisms speculated to have a central role in human adaptation to high altitude, connecting environmental, physiological, and genomic factors, in short periods [33, 35], it has been suggested that genetic adaptation requires longer periods to produce resistance to higher altitudes [15, 33]. Therefore, we believe that missing ancestry information does not imply a significant source of bias in our findings.

Due to the lack of information, we could not assess *(i)* distal factors, such as cultural behaviors or environmental exposures, *(ii)* intermediate factors, like maternal educational level, coverage of maternal health; or, *(iii)* proximal factors, like ethnicity, genetic or physiological adaptations. To our understanding, our main sensitivity analysis considers the role of the available confounders at individual level, and several contextual factors; therefore, this study lacks sources of significant bias. We encourage further research to clarify the role and specific mechanisms of how those factors affect neonatal outcomes.

## Conclusion

Higher altitudes are independently associated with shorter survival time, as measured by days, among at-risk neonates. Individual and contextual variables, such as altitude, should be considered when assessing the risk of neonates, especially in the highlands.

## Supporting information

**S1 Data.**
(CSV)

**S1 File. Material and methods complete version.**
(DOCX)

**S1 Fig. Diagram which shows the number of neonates registered, excluded, and included in the analysis.**
(TIF)

**S2 Fig. Histogram of the distribution of neonatal deaths per each one of the intervals of altitude of the health care facility where neonates were attended.**
(TIF)

**S3 Fig. Quantile-Normal plots by the transformation of altitude of the health care facility where neonates were attended.**
(TIF)

**S1 Table. Crude and adjusted neonatal mortality adjusted hazard ratios per each altitude stratum according to mixed-effects multivariate Cox proportional hazards models.**
(DOCX)

**S2 Table. Individual and contextual variables of the study population across categories of altitude of the health facility where neonates were attended.**
(DOCX)

**S3 Table. Neonatal mortality adjusted hazard ratios per each altitude stratum according to three mixed-effects multivariate Cox proportional hazards models.**
(DOCX)

**S4 Table. Crude and adjusted hazard ratios of the incidence rate of death per each altitude stratum, estimated by the final (parsimonious) mixed-effects Cox proportional hazards model, excluding neonates who died from asphyxia related disorders.**
(DOCX)

**S5 Table. Crude and adjusted hazard ratios of the incidence rate of death per each altitude stratum, estimated by the final (parsimonious) fixed effects Cox proportional hazards model, excluding neonates who died from congenital malformations.**
(DOCX)

**S6 Table. Crude and adjusted hazard ratios of the incidence rate of death per each altitude stratum, estimated by the final (parsimonious) fixed effects Cox proportional hazards model, excluding neonates who died from prematurity related disorders.**
(DOCX)

**S7 Table. Crude and adjusted hazard ratios of the incidence rate of death per each altitude stratum, estimated by the final (parsimonious) fixed effects Cox proportional hazards model, excluding neonates who died from infectious disorders.**
(DOCX)

**S8 Table. Crude and adjusted hazard ratios of the incidence rate of death per each altitude stratum, estimated by the final (parsimonious) fixed effects Cox proportional hazards model, excluding neonates who died from those with Apgar at 5 minutes $<$5.**
(DOCX)

**S9 Table. Crude and adjusted hazard ratios of the incidence rate of death per each altitude stratum, estimated by the final (parsimonious) fixed effects Cox proportional hazards model, excluding neonates who died from those with Apgar at 5 minutes $\geq$7.**
(DOCX)

**S10 Table. Crude and adjusted hazard ratios of the incidence rate of death per each altitude stratum, estimated by the final (parsimonious) fixed effects Cox proportional hazards model, excluding neonates who died outside their mother's usual residence province.**
(DOCX)

**S11 Table. Neonatal mortality adjusted hazard ratios per each altitude stratum according to two mixed-effects multivariate Cox proportional hazards models in which the mean Gini coefficient at the provincial level from 2014 to 2016 is considered as a contextual variable.**
(DOCX)

**S12 Table. Neonatal mortality adjusted hazard ratios per each altitude stratum according to two mixed-effects multivariate Cox proportional hazards models in which rural-urban health care is considered as a contextual variable.**
(DOCX)

**S13 Table. Adjusted odds ratios of death <24h, <48h, <72h, and <7 days of life per each altitude stratum according to four mixed-effects multivariate logistic regression models adjusted by the covariates of the model 1 (see main text and S3 Table).**
(DOCX)

**S14 Table. Adjusted estimations of survival in days of life across altitudes by Inverse-probability-weighted estimators (IPW) [5] using the covariates of the Model 1 (see main text and S3 Table) at different artificial cut-offs of the follow up.**
(DOCX)

## Acknowledgments

Authors thank the input from the "Score Bebé" project members. We, also, acknowledge the contributions from the Ministry of Public Health (*Ministerio de Salud Pública del Ecuador*, *MSP*). We would like to thank the input provided by the technical persons from Ministry of Public Health contributing to the project lifetime.

## Author Contributions

**Conceptualization:** Iván Dueñas-Espín, Luciana Armijos-Acurio, Estefanía Espín, Fernando Espinosa-Herrera, Ruth Jimbo, Ángela León-Cáceres, Raif Nasre-Nasser, María F. Rivadeneira, David Rojas-Rueda, Laura Ruiz-Cedeño, Betzabé Tello, Daniela Vásconez-Romero.

**Data curation:** Iván Dueñas-Espín, Luciana Armijos-Acurio, Raif Nasre-Nasser, Laura Ruiz-Cedeño, Daniela Vásconez-Romero.

**Formal analysis:** Iván Dueñas-Espín.

**Funding acquisition:** Iván Dueñas-Espín, Luciana Armijos-Acurio, Ruth Jimbo, Betzabé Tello.

**Investigation:** Iván Dueñas-Espín.

**Methodology:** Iván Dueñas-Espín, Luciana Armijos-Acurio, María F. Rivadeneira.

**Project administration:** Iván Dueñas-Espín.

**Validation:** Iván Dueñas-Espín.

**Writing – original draft:** Iván Dueñas-Espín, Luciana Armijos-Acurio, Ángela León-Cáceres.

**Writing – review & editing:** Iván Dueñas-Espín, Luciana Armijos-Acurio, Estefanía Espín, Fernando Espinosa-Herrera, Ruth Jimbo, Ángela León-Cáceres, Raif Nasre-Nasser, María F. Rivadeneira, David Rojas-Rueda, Laura Ruiz-Cedeño, Betzabé Tello, Daniela Vásconez-Romero.

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
