## [Decision Letter · Decision Letter 0]

20 Apr 2021

PONE-D-21-07809

Is a higher altitude associated with shorter survival among at-risk neonates?

PLOS ONE

Dear Dr. Dueñas Espin,

Thank you for submitting your manuscript to PLOS ONE. After careful consideration, we feel that it has merit but does not fully meet PLOS ONE’s publication criteria as it currently stands. Therefore, we invite you to submit a revised version of the manuscript that addresses the points raised during the review process.

Please follow the reviewers advice carefully.

We look forward to receiving your revised manuscript.

Kind regards,

Kazumichi Fujioka

Academic Editor

PLOS ONE

Journal Requirements:

3. Please refer to the specific statistical analyses performed as well as any post-hoc corrections to correct for multiple comparisons. If these were not performed please justify the reasons. Please refer to our statistical reporting guidelines for assistance (https://journals.plos.org/plosone/s/submission-guidelines.#loc-statistical-reporting).

4. One of the noted authors is a group or consortium [“Score Bebé” project]. In addition to naming the author group, please list the individual authors and affiliations within this group in the acknowledgments section of your manuscript. Please also indicate clearly a lead author for this group along with a contact email address.

Reviewers' comments:

Reviewer's Responses to Questions

**Comments to the Author**

1. Is the manuscript technically sound, and do the data support the conclusions?

Reviewer #1: Yes

Reviewer #2: Yes

2. Has the statistical analysis been performed appropriately and rigorously? 

Reviewer #1: Yes

Reviewer #2: I Don't Know

3. Have the authors made all data underlying the findings in their manuscript fully available?

Reviewer #1: Yes

Reviewer #2: Yes

4. Is the manuscript presented in an intelligible fashion and written in standard English?

Reviewer #1: Yes

Reviewer #2: Yes

5. Review Comments to the Author

Reviewer #1: General Comments:

This paper showed that high altitude is a risk factor for neonatal mortality. There are some weaknesses in this paper, such as the lack of data on surviving neonates and the small number of facilities at elevations of 80-2750 meters. However, the results of this study are of higher quality than existing reports and provide significant evidence for an association between elevation and risk of neonatal death.

Specific recommendations for minor revision: It seems to me that the intent of the research is difficult to convey. By shortening the introduction slightly and changing the order of the sentences, the intent of the study would be clearer.

1) line 64-84: How about changing it as follows?

Scientific evidence has consistently shown that high altitude is associated with high neonatal mortality [9]. Specifically, in the prenatal phase, a pregnancy at high altitude has lower availability of iron, vitamins A and D, iodine [1], higher erythropoietic demands, reduced blood oxygen content [9], and a subsequent lower maternal oxygen delivery to the fetoplacental unit. These mechanisms could result in lower foetal growth [9] and in a decrease in placental weight, accompanied by a significant increase in the incidence of syncytial nodes, trophoblastic cells, and foetal capillaries with a large intervillous space and fewer villi. The disadvantages in oxygen diffusion can also contribute to low birth weight in full-term neonates, preeclampsia, and gestational hypertension [22]. Hence, the transition from oxygenation through the placenta to oxygenation through the lungs is poor in high altitudes [9] the transition period for neonates in high altitudes is longer; it means that, even in the absence of pulmonary disorders, neonates may experience arterial oxygen desaturation during the first week of life [9].

2) line 85-89: I suggest that you remove this content from the introduction. It should be mentioned in the discussion.

3) line 90-94: I propose the following change.

However, past studies have an ecological approach [10–13], small sample sizes [14], lack of consideration of individual confounders in the analyses (i.e., birth weight, gestational age, small for gestational age, Apgar scale, type of delivery, and the presence of specific comorbidities) [14], lack of contextual factors (i.e., differences in quality of care, type of health care facility [i.e., public or private], and level of care [15–17], and most of them only compared two or three altitude categories [10–14,18–20]. Despite the available conceptual model of stillbirth health determinants [21], this model does not include altitude as a determinant. This background represents an evidence-based conceptual framework that suggests that neonatal survival could be negatively associated with high altitude, especially among at-risk neonates. The current study aims to assess such association by setting survival among at risk neonates, as measured by days, as the primary outcome, using a nationwide information database.

4) line 304-306: This expression is difficult to understand as a limitation. I suggest you change it as follows.

Second, the present study had a lack of data on individual socioeconomic factors –a strong neonatal health determinant– [3]. However, we believe that the aggregated data used in this analysis could capture such socioeconomic effects, resulting in robust estimations.

5) line 307-315: We recommend the following change.

Also, when altitude is considered as a potential health determinant in the neonatal period, it is required to consider that the mother-baby binomial could carry genetic traits that are not well suited to their current environment. This might define their degree of adaptation to high altitude. Thus, affect their susceptibility to maladaptation problems that could increase mortality risk [2,23]. Neonatal mortality at high altitude in South America is reduced by maternal and paternal indigenous Andean ancestry, in a dose-dependent manner [33,34], mechanisms and genes related to the ancestry protective effect are related to enhanced uteroplacental blood flow [23,33,35,36]. Additionally, despite epigenetic mechanisms speculated to have a central role in human adaptation to high altitude, connecting environmental, physiological, and genomic factors, in short periods [33,35], it has been postulated that genetic adaptation requires longer periods to produce resistance to higher altitudes [14,33]. Therefore, we believe that missing ancestry information does not imply a significant source of bias in our findings.

Reviewer #2: The manuscript presents a retrospective analysis of Ecuadorian neonates who died at≦28days of life, and assess the association between higher altitudes on survival time among at-risk neonates. The study concludes that higher altitudes are independently with shorter survival time. The following comments and suggestions will improve the manuscript:

1. Introduction: “fetal” is more common than “foetal” (L76, 78..), so you should change.

2. Result: you state that “Except for the percentage of small for gestational age, there were

significant differences between the individual and contextual variables across altitude

categories (Table 1).” Why are there significant differences in the background (the individual and contextual variables) at different altitudes? Please discuss about this.

3. Result: the altitude was categorized two patterns, (ⅰ)＜10m, between 10 to ＜2500m, and ≧2500m, （ⅱ）＜80m, between 80 to <2500m, from 2500 to <2750m, and ≧2750m. Please clarify why you have categorized in this way, and list any previous literature you have used as reference.

4.Result: you state that“（ⅱ）considering being attended at＜80m as the reference, being attended between 80 to <2500m, from 2500 to <2750m, and ≧2750m there was 42%, 57% and 17% higher HR, respectively.”(page12, Line 220-222) Why do you think the HR of ≥2750m is the lowest?

6. PLOS authors have the option to publish the peer review history of their article (what does this mean?). If published, this will include your full peer review and any attached files.

Reviewer #1: **Yes: **Mitsuhiro Okamoto

Reviewer #2: No

---

## [Author Response · Author response to Decision Letter 0]

31 May 2021

Rebuttal letter to reviewers’ comments to authors

PLOS ONE 

Manuscript ID: PONE-D-21-07809

Title: Is a higher altitude associated with shorter survival among at-risk neonates?

Iván Dueñas-Espín, Luciana Armijos-Acurio, Estefanía Espín, Fernando Espinosa-Herrera, Ruth Jimbo, Ángela León-Cáceres, Raif Nasre-Nasser, María F. Rivadeneira, David Rojas-Rueda, Laura Ruiz-Cedeño, Betzabé Tello, Daniela Vásconez-Romero.

Decision: Minor revision

Article Type: original article.

Corresponding Author: Iván Dueñas-Espín

We thank the Editor and Reviewers for their insightful comments that have been helpful to improve the manuscript. It is very much appreciated. While it is known that living at higher altitudes can pose health risks, the specific outcomes on neonatal health outcomes are unknown. We found that higher altitudes were independently associated with shorter survival time, as measured by days among at-risk neonates. 

JOURNAL REQUIREMENTS:

Please review your reference list to ensure that it is complete and correct. If you have cited papers that have been retracted, please include the rationale for doing so in the manuscript text or remove these references and replace them with relevant current references. Any changes to the reference list should be mentioned in the rebuttal letter that accompanies your revised manuscript. If you need to cite a retracted article, indicate the article’s retracted status in the References list and also include a citation and full reference for the retraction notice.

Thanks a lot for your accurate comment. We have carefully reviewed all the references and we found that one reference was retracted: 

American College of Obstetricians. ACOG Practice bulletin no. 134: fetal growth restriction. Obstet Gynecol 2013;121:1122–33.

After this, we have replaced it with a relevant current reference. The new reference is the number 24:

24. Damhuis SE, Ganzevoort W, Gordijn SJ. Abnormal Fetal Growth Small for Gestational Age, Fetal Growth Restriction, Large for Gestational Age: Definitions and Epidemiology. Obstet Gynecol Clin NA [Internet]. 2021;48(2):267–79. Available from: https://doi.org/10.1016/j.ogc.2021.02.002

Thank you for your suggestion. We have revised the format in the templates and have changed the file names of the supporting information files in both, the main manuscript and in the file names, respectively.

 2. Please provide additional details regarding participant consent. In the ethics statement in the Methods and online submission information, please ensure that you have specified (1) whether consent was informed and (2) what type you obtained (for instance, written or verbal, and if verbal, how it was documented and witnessed). If your study included minors, state whether you obtained consent from parents or guardians. If the need for consent was waived by the ethics committee, please include this information. If you are reporting a retrospective study of medical records or archived samples, please ensure that you have discussed whether all data were fully anonymized before you accessed them and/or whether the IRB or ethics committee waived the requirement for informed consent. If patients provided informed written consent to have data from their medical records used in research, please include this information.

Thank you for your observations. We have added a paragraph entitled “Ethics approval” in the last part of the Methods section (p. 11, L231 of the marked-up copy), as follows:

“Ethics approval

This study was part of the “Score Bebé” project and was conducted with the authorization of the Research Ethics Committee in Human Beings (CEISH) of the Pontificia Universidad Católica del Ecuador approval (code number: 2018-09-EO). The need for consent was waived by the ethics committee given that the study was based on secondary information; therefore, informed consent was not obtained. All data were fully anonymized before accessed.” 

3. Please refer to the specific statistical analyses performed as well as any post-hoc corrections to correct for multiple comparisons. If these were not performed please justify the reasons. Please refer to our statistical reporting guidelines for assistance (https://journals.plos.org/plosone/s/submission-guidelines.#loc-statistical-reporting).

Thank you very much. We have added the following text in the methods section (p. 9, L 177 of the marked-up copy):

“(ii) for assessing differences in birth weight we used one-way ANOVA and Tukey post hoc test, …”

Additionally, we have clarified it in the Table 1 caption.

4. One of the noted authors is a group or consortium [“Score Bebé” project]. In addition to naming the author group, please list the individual authors and affiliations within this group in the acknowledgments section of your manuscript. Please also indicate clearly a lead author for this group along with a contact email address.

Thank you. After a brief discussion among the group, we decided to not include the group as another author. The reason is that the total members of the group participated as authors in the manuscript; therefore, the consortium name in the author’s list is not necessary.

COMMENTS TO THE AUTHOR

1. Is the manuscript technically sound, and do the data support the conclusions?

Reviewer #1: Yes

Reviewer #2: Yes

Thank you.

2. Has the statistical analysis been performed appropriately and rigorously?

Reviewer #1: Yes

Reviewer #2: I Don't Know

Thank you. We have revised the specific statistical analyses performed as well as any post-hoc corrections to correct for multiple comparisons as the reporting guidelines states. We have added the following text in the methods section (p. 9, L 177 of the marked-up copy):

“(ii) for assessing differences in birth weight we used one-way ANOVA and Tukey post hoc test, …”

Additionally, we have clarified it in the Table 1 caption.

3. Have the authors made all data underlying the findings in their manuscript fully available?

Reviewer #1: Yes

Reviewer #2: Yes

Thank you.

4. Is the manuscript presented in an intelligible fashion and written in standard English?

Reviewer #1: Yes

Reviewer #2: Yes

Thank you.

5. Review Comments to the Author

ANSWERS TO REVIEWER 1

Reviewer #1: General Comments:

This paper showed that high altitude is a risk factor for neonatal mortality. There are some weaknesses in this paper, such as the lack of data on surviving neonates and the small number of facilities at elevations of 80-2750 meters. However, the results of this study are of higher quality than existing reports and provide significant evidence for an association between elevation and risk of neonatal death.

Thank you. We agree with the comment of the reviewer announcing that there are some weaknesses in the manuscript. We have tried to take advantage of the available information to demonstrate that high altitude is a risk factor for neonatal mortality by an individual based analysis of the nationwide database of neonatal mortality in Ecuador. Despite those weaknesses we think our manuscript supports such association by rigorous scientific evidence.

Specific recommendations for minor revision: It seems to me that the intent of the research is difficult to convey. By shortening the introduction slightly and changing the order of the sentences, the intent of the study would be clearer. 

Thank you very much for your insightful suggestion. We have accepted your suggestion of change, as follows: 

1) line 64-84 (current lines 64 to 68 of the of the marked-up copy):

“Scientific evidence has consistently shown that high altitude is associated with high neonatal mortality [9]. Specifically, in the prenatal phase, a pregnancy at high altitude has lower availability of iron, vitamins A and D, iodine [1], higher erythropoietic demands, reduced blood oxygen content [9], and a subsequent lower maternal oxygen delivery to the fetoplacental unit. These mechanisms could result in lower fetal growth [9]. and in a decrease in placental weight, accompanied by a significant increase in the incidence of syncytial nodes, trophoblastic cells, and fetal capillaries with a large intervillous space and fewer villi. The disadvantages in oxygen diffusion can also contribute to low birth weight in full-term neonates, preeclampsia, and gestational hypertension [22]. Hence, the transition from oxygenation through the placenta to oxygenation through the lungs is poor in high altitudes [9] the transition period for neonates in high altitudes is longer; it means that, even in the absence of pulmonary disorders, neonates may experience arterial oxygen desaturation during the first week of life [9].” 

References:

1. Huddleston B, de Salvo EAP, Zanetti M, Bloise M, Bel J, Franceschini G, et al. Towards a GIS-based analysis of mountain environments and populations [Internet]. Environment and Natural Resources working paper No. 10. 2003 [cited 2019 Oct 24]. Available from: http://www.fao.org/family-farming/detail/en/c/285416/

9. Niermeyer S, Mollinedo PA, Huicho L. Child health and living at high altitude. Arch Dis Child. 2009;94(10):806–11.

22. Gassmann NN, Van Elteren HA, Goos TG, Morales CR, Rivera-Ch M, Martin DS, et al. Pregnancy at high altitude in the Andes leads to increased total vessel density in healthy newborns. J Appl Physiol. 121(3):709–15.

2) line 85-89: I suggest that you remove this content from the introduction. It should be mentioned in the discussion.

Thank you, It was done.

3) line 90-94 (current lines 104 to 115 of the marked-up copy):

Thank you very much. We have accepted your suggestion of change, and we proceeded as follows: 

“However, past studies have an ecological approach [10–13], small sample sizes [14], lack of consideration of individual confounders in the analyses (i.e., birth weight, gestational age, small for gestational age, Apgar scale, type of delivery, and the presence of specific comorbidities) [14], lack of contextual factors (i.e., differences in quality of care, type of health care facility [i.e., public or private], and level of care [15–17], and most of them only compared two or three altitude categories [10–14,18–20]. Despite the available conceptual model of stillbirth health determinants [21], this model does not include altitude as a determinant. This background represents an evidence-based conceptual framework that suggests that neonatal survival could be negatively associated with high altitude, especially among at-risk neonates. The current study aims to assess such association by setting survival among at-risk neonates, as measured by days, as the primary outcome, using a nationwide information database.”

References:

10. Unger C, Weiser JK, McCullough RE, Keefer S, Moore LG. Altitude, low birth weight, and infant mortality in colorado. Obstet Gynecol Surv. 1989;44(4):253–4. 

11. Mazess RB. Neonatal mortality and altitude in Peru. Am J Phys Anthropol. 1965;23(3):209–13. 

12. Frisancho AR, Cossman J. Secular trend in neonatal mortality in the mountain states. Am J Phys Anthropol. 1970;33(1):103–5. 

13. Grahn D, Kratchman J. Variation in Neonatal Death Rate and Birth Weight in the United States and Possible Relations To Environmental Radiation, Geology and Altitude. Am J Hum Genet. 1963;15:329–52. 

14. Wiley AS. Neonatal size and infant mortality at high altitude in the western Himalaya. Am J Phys Anthropol [Internet]. 1994;94(3):289–305. Available from: http://ovidsp.ovid.com/ovidweb.cgi?T=JS&PAGE=reference&D=emed3&NEWS=N&AN=1994204195

15. Opondo C, Allen E, Todd J, English M. Association of the Paediatric Admission Quality of Care score with mortality in Kenyan hospitals: a validation study. Lancet Glob Heal [Internet]. 2018;6(2):e203–10. Available from: http://dx.doi.org/10.1016/S2214-109X(17)30484-9

16. Adams N, Tudehope D, Gibbons KS, Flenady V. Perinatal mortality disparities between public care and private obstetrician-led care: a propensity score analysis. BJOG An Int J Obstet Gynaecol. 2018;125(2):149–58. 

17. Lassi ZS, Kumar R, Mansoor T, Salam RA, Das JK, Bhutta ZA. Essential interventions: Implementation strategies and proposed packages of care. Reprod Health [Internet]. 2014;11(Suppl 1):S5. Available from: http://www.reproductive-health-journal.com/content/11/S1/S5

18. Levine RS, Salemi JL, Mejia De Grubb MC, Wood SK, Gittner L, Khan H, et al. Altitude and Variable Effects on Infant Mortality in the United States. High Alt Med Biol. 2018;19(3):265–71. 

19. Rothhammer F, Fuentes-Guajardo M, Chakraborty R, Bermejo JL, Dittmar M. Neonatal variables, altitude of residence and Aymara Ancestry in Northern Chile. PLoS One. 2015;10(4):1–10. 

20. Beall CM. Optimal birthweights in Peruvian populations at high and low altitudes. Am J Phys Anthropol. 1981;56(3):209–16.

21. World Health Organization. Closing the gap in a generation: Health equity through action on the social determinants of Health. 2008.

4) line 304-306:

Thank you very much. We have accepted your suggestion , and we changed the seventh paragraph of the discussion (current lines 348 to 350 of the marked-up copy), as follows:

“Second, the present study had a lack of data on individual socioeconomic factors –a strong neonatal health determinant– [3]. However, we believe that the aggregated data used in this analysis could capture such socioeconomic effects, resulting in robust estimations.”

Reference:

3. Liu L, Oza S, Hogan D, Chu Y, Perin J, Zhu J, et al. Global, regional, and national causes of under-5 mortality in 2000–15: an updated systematic analysis with implications for the Sustainable Development Goals. Lancet [Internet]. 2016;388(10063):3027–35. Available from: http://dx.doi.org/10.1016/S0140-6736(16)31593-8

5) line 307-315: 

Thank you so much. We have accepted your suggestion and we changed the eight paragraph of the discussion (current lines 354 to 366 of the marked-up copy), as follows:

“Also, when altitude is considered as a potential health determinant in the neonatal period, it is required to consider that the mother-baby binomial could carry genetic traits that are not well suited to their current environment. This might define their degree of adaptation to high altitude. Thus, affect their susceptibility to maladaptation problems that could increase mortality risk [2,23]. Neonatal mortality at high altitude in South America is reduced by maternal and paternal indigenous Andean ancestry, in a dose-dependent manner [33,34], mechanisms and genes related to the ancestry protective effect are related to enhanced uteroplacental blood flow [23,33,35,36]. Additionally, despite epigenetic mechanisms speculated to have a central role in human adaptation to high altitude, connecting environmental, physiological, and genomic factors, in short periods [33,35], it has been suggested that genetic adaptation requires longer periods to produce resistance to higher altitudes [14,33]. Therefore, we believe that missing ancestry information does not imply a significant source of bias in our findings.”

Reference:

2. Wiley AS. An Ecology of High-Altitude Infancy. An Ecology of High-Altitude Infancy. Cambridge University Press; 2004.

14. Wiley AS. Neonatal size and infant mortality at high altitude in the western Himalaya. Am J Phys Anthropol [Internet]. 1994;94(3):289–305. Available from: http://ovidsp.ovid.com/ovidweb.cgi?T=JS&PAGE=reference&D=emed3&NEWS=N&AN=1994204195

23. Julian CG, Wilson MJ, Moore LG. Evolutionary adaptation to high altitude: A view from in utero. Am J Hum Biol. 2009;21(5):614–22.

33. Julian CG, Moore LG. Human genetic adaptation to high altitude: Evidence from the andes. Genes (Basel). 2019;10(2). 

34. 34. Bennett A, Sain SR, Vargas E, Moore LG. Evidence that parent-of-origin affects birth-weight reductions at high altitude. Am J Hum Biol. 2008;20(5):592–7. 

35. 35. Julian CG. Epigenomics and human adaptation to high altitude. J Appl Physiol. 2017;123(5):1362–70. 

36. 36. Van Patot MCT, Murray AJ, Beckey V, Cindrova-Davies T, Johns J, Zwerdlinger L, et al. Human placental metabolic adaptation to chronic hypoxia, high altitude: Hypoxic preconditioning. Am J Physiol - Regul Integr Comp Physiol. 2010;298(1):166–73.

ANSWERS TO REVIEWER 2

Reviewer #2: Comments:

The manuscript presents a retrospective analysis of Ecuadorian neonates who died at≦28days of life, and assess the association between higher altitudes on survival time among at-risk neonates. The study concludes that higher altitudes are independently with shorter survival time. The following comments and suggestions will improve the manuscript:

1. Introduction: “fetal” is more common than “foetal” (L76, 78..), so you should change.

Thank you very much. We agree with the reviewer on the accurate observation. After a careful review, we corrected the word in the whole manuscript.

2. Result: you state that “Except for the percentage of small for gestational age, there were

significant differences between the individual and contextual variables across altitude

categories (Table 1).” Why are there significant differences in the background (the individual and contextual variables) at different altitudes? Please discuss about this.

Thank you so much for your insightful comment. We agree that it needs further explanation. In this regard, we have added additional explanation in the discussion section (L309 of the marked-up copy), as follows:

 "Second, as previous evidence has suggested, altitude is not only a physical characteristic, but it is composed of complex social, environmental, and healthcare characteristics. For example, Pichincha –the province where the Ecuadorian capital Quito is located (2850 m)– has more health services and public practice physicians per capita than any other province in the country [29]. This could partially explain the reduced HR compared with lesser altitudes (but not with the lowest altitude). Moreover, our results showed that individual and contextual variables were significantly different across altitudes strata. This supports the statement that altitude is not only a challenge in genetic, epigenetic, and physiologic domains, but a complex interaction between the environmental, social, and biological domains."

Reference:

29. López-Cevallos DF, Chi C. Health care utilization in Ecuador: A multilevel analysis of socio-economic determinants and inequality issues. Health Policy Plan. 2010;25(3):209–18.

3. Result: the altitude was categorized two patterns, (ⅰ)＜10m, between 10 to ＜2500m, and ≧2500m, （ⅱ）＜80m, between 80 to <2500m, from 2500 to <2750m, and ≧2750m. Please clarify why you have categorized in this way, and list any previous literature you have used as reference.

Thank you so much for your insightful comment. In this regard, we have added additional explanation in the S1 Materials and methods complete version, as follows:

“In order to perform a proper categorization of the altitude we considered several geographic factors. Ecuador has four regions, each with different altitude and specific distribution of the population. The coast region and the Galapagos Islands region are located at an altitude that varies among 0–500 m. The highlands, include the Andean mountains, located at beyond 1501 m. The Amazon region is located between 501–1500 m [2]. The diversity of climates and altitude levels, the political division of the country, and the limited access to basic services, determine a particular configuration of the population´s density. Consequently, regarding the evaluated health outcome, distinct studies established different altitude clusters. For example, a study of the prevalence of metabolic syndrome in Ecuador, used the above-mentioned categories [2]. Contrary, a study about congenital heart disease in Ecuador, determined three altitude clusters according to CHD prevalence: 2500 to 2750 m; 2751 to 3000 m; and 3001 and 3264 m [3]. The last ones are similar to our altitude clusters, but noticeably they do not include low altitude clusters since the prevalence of CHD is not significant in those regions. Thus, the patterns of altitude categorization that we used, are according to our research objective.” 

References

2. Pérez-Galarza J, Baldeón L, Franco OH, Muka T, Drexhage HA, Voortman T, et al. Prevalence of overweight and metabolic syndrome, and associated sociodemographic factors among adult Ecuadorian populations: the ENSANUT-ECU study. J Endocrinol Invest [Internet]. 2021;44(1):63–74. Available from: https://doi.org/10.1007/s40618-020-01267-9

3. González-Andrade F. High Altitude as a Cause of Congenital Heart Defects: A Medical Hypothesis Rediscovered in Ecuador. High Alt Med Biol. 2020;21(2):126–34.

Also, we have added next text in Methods section (current lines 148 to 154 of the marked-up copy):

“In order to perform a proper categorization of the altitude we considered that Ecuador has four regions, each with different altitudes and a specific distribution of the population [24]. Nevertheless, regarding the health outcome evaluated, distinct studies established different altitude clusters. Thus, the patterns of altitude categorization we used, are according to our research objective.”

Reference:

24. Pérez-Galarza J, Baldeón L, Franco OH, Muka T, Drexhage HA, Voortman T, et al. Prevalence of overweight and metabolic syndrome, and associated sociodemographic factors among adult Ecuadorian populations: the ENSANUT-ECU study. J Endocrinol Invest [Internet]. 2021;44(1):63–74. Available from: https://doi.org/10.1007/s40618-020-01267-9

4.Result: you state that“（ⅱ）considering being attended at＜80m as the reference, being attended between 80 to <2500m, from 2500 to <2750m, and ≧2750m there was 42%, 57% and 17% higher HR, respectively.”(page12, Line 220-222) Why do you think the HR of ≥2750m is the lowest?

Thank you for your observation. As we found an inverted J shape, with an apparent “improvement” in the survival at a certain altitude (>2750 m) when comparing to lower altitudes, but not to the lowest, during fixed effects modelling estimations, several considerations could preclude us from establishing such a conclusion. This has been written in the Discussion section (p.15 last paragraph, lines 304 to 318 of the marked-up copy) as follows: 

“First, when accounting for the data clustering (of administrative planning areas, type of health care facility, and level of care), the random-effects modelling provided a useful approach for simultaneously estimating the parameters of the regression model and the variance components that account for the contextual factors. It resulted in a more realistic linear shape association. Second, as previous evidence has suggested, altitude is not only a physical characteristic, but it is composed of complex social, environmental, and healthcare characteristics. For example, Pichincha –the province where the Ecuadorian capital Quito is located (2850 m)– has more health services and public practice physicians per capita than any other province in the country. This could partially explain the reduced HR compared with lesser altitudes (but not with the lowest altitude). We believe that the association between altitude and survival is more like a linear shape and that the inverted J shape hypothesis is a relevant issue to explore, but it goes beyond the scope of this study.”

OTHER JOURNAL REQUIREMENTS

Thank you very much. We have checked the Figure 1 and it has been properly modified according to journals’ requirements. The Preflight Analysis and Conversion Engine (PACE) digital diagnostic tool was very useful to identify the errors. Current version includes the modified Figure 1.

---

## [Editor Report · Decision Letter 1]

7 Jun 2021

Is a higher altitude associated with shorter survival among at-risk neonates?

PONE-D-21-07809R1

Dear Dr. Dueñas Espin,

We’re pleased to inform you that your manuscript has been judged scientifically suitable for publication and will be formally accepted for publication once it meets all outstanding technical requirements.

Kind regards,

Kazumichi Fujioka

Academic Editor

PLOS ONE

Additional Editor Comments (optional):

Sufficiently revised following reviewers advice.
---

## [Editor Report · Acceptance letter]

23 Jun 2021

PONE-D-21-07809R1 

Is a higher altitude associated with shorter survival among at-risk neonates? 

Dear Dr. Dueñas-Espin:

I'm pleased to inform you that your manuscript has been deemed suitable for publication in PLOS ONE. Congratulations! Your manuscript is now with our production department. 

Kind regards, 

on behalf of

Dr. Kazumichi Fujioka 

Academic Editor

PLOS ONE